# Young Adults' Perception of Breadcrumbing Victimization in Dating Relationships

**Vivek Khattar [1],\*, Shreya Upadhyay [1] and Raúl Navarro [2]**

[1]   Faculty of Education & Psychology, The Maharaja Sayajirao University of Baroda, Vadodara 390002, India
[2]   Faculty of Education and Humanities, University of Castilla-La Mancha, 16071 Cuenca, Spain
\*    Correspondence: vivekkhattar37@gmail.com

**Abstract:** Background: Breadcrumbing is an unexplored dating trend disguised in the form of subtle manipulation in relationships. With the increase in online dating apps, people have started to initiate, maintain and end relationships, and the use of manipulative tactics have increased on such platforms. The present study explores the meaning of breadcrumbing and its effects on the breadcrumbie's mental health and wellbeing. Method: The research design was qualitative in nature through the use of focus group discussions. Two focus group discussions were conducted including nine participants in total (one male, eight females). Results: After the data analysis, five major themes emerged defining breadcrumbing—charm, leading on, incongruence, avoiding emotional investment and commitment uncertainty. Conversational fragments also revealed that breadcrumbing had an impact on the breadcrumbie's future relationships, emotional disturbance, self-concept, and signs of depression. The red flags and effective coping strategies were also discovered with the help of a thematic analysis. Further research on personality correlates of breadcrumbing perpetration and victimization is recommended.

**Keywords:** breadcrumbing; manipulation; dating; emotional abuse; young adults





## 1. Introduction

Dating and romantic relationships are an important aspect of young people's lives. Dating simply refers to the initial stage of forming romantic ties with a person to assess their suitability as a prospective partner [1]. Although in recent years the connotations of dating have become ambiguous, loosely referred to as "getting to know each other", it also means spending time with someone you find attractive in the hope of building a committed relationship [2,3].

There is an increasing body of evidence showing that the use of dating apps to meet new people is significantly growing [4], especially in India [5]. The dating trend began in India in the late 1990s with the parent company Matrimony.com, followed by Jeevansathi.com and Shaadi.com offering two types of services, that is, a marriage service and matchmaking service. With the help of these websites, individuals used to create a profile, register on the website and wait for the suitable match with the purpose of getting married. However, in today's age, with the growth of casual dating in India, dating apps such as Tinder, Bumble, OkCupid, TrulyMadly, Happn, QuackQuack and so on have outgrown traditional matrimonial websites. These apps provide services similar to the matrimonial sites. Dating apps are growing rapidly in small towns and cities, making India the second-largest revenue market for dating apps after the USA [6].

Through various dating apps, dating seems to be a successful venture to meet new people and form relationships quickly and effectively. Due to the easy access to these platforms, some drawbacks have also been observed, such as victimization, sexual and impulsive behavior, fear of deception, objectification, gamification of relationships, cyber-stalking and other behaviors such as benching breadcrumbing, ghosting, haunting, orbiting

and slow-fading [7–9]. These behaviors are the examples of how individuals utilize dating applications to seduce, initiate, maintain, manipulate and end relationships. However, there are only a few studies exploring these online behaviors, and specifically research analyzing how different forms of manipulation such as breadcrumbing within the romantic relationship impact on individuals' mental health is scarce [10].

Breadcrumbing is one of the forms of manipulation defined as "the act of sending out flirtatious, but non-committal text messages (i.e., breadcrumbs) in order to lure a sexual/romantic partner without expending much effort" [11]. It has also been defined as "a euphemism for leading someone on by contacting them intermittently to keep the other person interested" [12], and "sporadically sending someone flirtatious yet non-committal text messages or random social media "likes" to keep the person's dating expectations of a possible relationship going, although the sender has no actual intentions of dating" [13]. In sum, breadcrumbing is a negative dating behavior that involves repeatedly tossing out just enough titbits of interest to keep another person interested and involved, where the breadcrumber is not truly interested in the person they are dating and is only using the relationship to gain a superficial connection and attention from them [14].

*Past Research on Breadcrumbing*

Although studies on behaviors such as ghosting are growing in recent years, there is still little empirical evidence about the new strategies used to manipulate or maintain relationships. For example, Navarro et al. [15] explored the prevalence of breadcrumbing among Spanish adults aged between 18 to 40 years. The findings revealed that breadcrumbing was a term familiar to only a half of the participants ($n = 626$). However, more than three in ten participants reported having encountered or even begun breadcrumbing in the previous 12 months. The results also revealed that the likelihood of experiencing as well as starting breadcrumbing increases with the usage of online dating services and apps. Breadcrumbing was a strategy used in short-term relationships characterized by less commitment. In this sense, they discussed that breadcrumbing could occur as a way to keep enjoying sporadic encounters while meeting other people or because breadcrumbers would like to keep the relationship casual or do not want to establish a monogamous relationship.

In a subsequent study, Navarro et al. [10] explored three psychological constructs including loneliness, satisfaction with life and helplessness and their relationships with breadcrumbing and ghosting experiences. Their findings showed that participants who reported experiencing breadcrumbing and combined forms (both breadcrumbing and ghosting) felt decreased life satisfaction, more helplessness, and self-perceived loneliness. To explain the association between breadcrumbing and the negative mental health correlates examined, they discussed that breadcrumbing may operate in a comparable direction to that of addictive behaviors. In this sense, the expectations generated by the breadcrumber (random communications via social networks), the anticipation of a possible reward for these behaviors (the possibility to meet in person), together with their lack of probability (it is not possible to predict when the reinforcement will be received), cause strain on those who suffer breadcrumbing and, in consequence, provoke negative psychological consequences.

Additionally, there has been a study conducted to design and psychometrically test Breadcrumbing in Affective-Sexual Relationships (BREAD-ASR)—a questionnaire investigating the practice of breadcrumbing among adolescents [16]. A paper-and-pencil survey was conducted in a high school in southeast Spain with 247 teenagers. Psychometric analysis for the test revealed satisfactory content and construct validity and good internal consistency. They found that breadcrumbing perpetration was characterized by sporadic communication, lack of commitment and avoidance of uncomfortable or negative interactions, but it was also characterized by dependence, given their necessity to reassert themselves and feeling ownership via having their victims available. They reported that the BREAD-ASR questionnaire is a valid and reliable tool that health professionals can use to screen for breadcrumbing perpetration and to create effective community prevention

and intervention programs that may assist and support young people and families in successfully navigating new forms of online relationships and perpetration. However, while qualitative studies have been already conducted to conceptualize and explore breakup strategies such as ghosting [17,18], as far as we know, no qualitative research has been developed to examine breadcrumbing experiences and conceptualize such phenomena. This study explores breadcrumbing experiences from a qualitative approach and seeks to understand the conceptual mechanisms of breadcrumbing behavior.

## 2. Materials and Methods

### 2.1. Research Design and Participants

As an exploratory study, focus group discussions (FGD) were conducted as a method of collecting data to gain an in-depth understanding of the participants' perspectives and experiences of breadcrumbing behavior in dating and relationships. Focus groups are a useful tool to explore topics about which very little is known and understood. They provided flexibility, freedom, and a permissive and non-threatening environment to share their views without the fear of judgment from other participants. The group environment enables a broad range of insights on the research topic in a single setting [19]. Therefore, focus group discussion was more appropriate for this study than the interview method.

In choosing the sample of participants, a snowball sampling method was used, which matched the purpose of the study. The inclusion criteria were based on participants who had experienced breadcrumbing in dating or relationships in the past one year. In undertaking this study, participants in the focus group discussion had to be aged between 18 and 35 years. The age restriction aimed to provide a more coherent group, which makes the experiences relatable to the participants. Participants were excluded from the study if they had only experienced behaviors such as ghosting, gaslighting or physical or sexual abuse in their relationships.

Participants were gathered through flyer distribution across all the batches of the Department of Psychology, [The Maharaja Sayajirao University], stating the purpose of the study and inclusion/exclusion criteria. We conducted the focused group discussions in two groups due to two reasons: the large number of participants and to moderate the group effectively. Group one consisted of five females (83.3%) and one male (16.7%), and group two consisted of six females (85.7%) and one male (14.3%) participants. However, two participants from group one left the discussion in the beginning as they had connectivity issues, and the researcher excluded two participants from group two as they did not meet the inclusion criteria. Therefore, nine participants took part in the study. The participants differed in several dimensions. The age range of the participants was 19–30 years (mean = 22.78 years). Eight participants were university students (undergraduate = four, postgraduate = four) and one participant was a self-employed professional. Four participants reported having no prior relationship experience, two participants had been in one long-term, committed relationship, and three participants had prior casual dating experience. This allowed the researcher to get a rich variety of perspectives on the topic.

### 2.2. Procedures

In the initial stage, in October 2020, a pilot was conducted through structured interviews with five individuals (three females and two males) who had experienced breadcrumbing in long-term, committed relationships. It was a telephonic interview, and all interviews were fully transcribed. The purpose of the pilot study was to have a better understanding of breadcrumbing in committed relationships and its effects on mental health. This process allowed the interviewee to express their thoughts and feelings relating to the questions. This opportunity allowed us to resolve any difficulties with the wordings of the questions and the structure, while also identifying some questions that might make a participant feel uncomfortable. The data collected in these interviews were not included in the current analysis. Based on the data from the pilot study, a final discussion guide was prepared.

The first group discussion was held in December 2020 over Skype video conferencing due to the COVID-19 pandemic restrictions. Six participants approached us in reference to the flyers that were sent across the college groups, out of which two of them had connectivity issues, so they left the discussion. Finally, there were three females and one male participant in the discussion. The researcher read out the instructions to the participants, and verbal and written consent was taken for the participation and recording of the session. The discussion guide was divided into three parts—introductory questions, interview questions and closing questions followed by the summary of the discussion (see Appendix A). The moderator uses the discussion guide as a resource to maintain the balance between the focus of the study and the group discussion [20]. The questions were mainly open ended, with a small number of closed ended questions relating to information such as where did they meet, the mode of communication in their relationship and their preference of the mode of communication. Some questions emerged during the group discussion, including, "What kind of hints did you receive from your partner?" and "How did you differentiate if your partner was being genuine or not?" The participants had the flexibility to use Hindi, Gujarati or English language to respond to the questions. After the discussion, the author fully transcribed the discussion. The second discussion was held in March 2021 with seven participants, from which the researcher excluded two of them as they did not fulfill the inclusion criteria. New questions were added in the discussion guide that emerged from the first group. The researcher did not initiate the third group discussion, as a lot of information was repetitive and reached data saturation. The participants did not receive any monetary reward or other form of compensation for participating in the study.

*2.3. Data Analysis*

After the focus group discussion, data were transcribed in a Microsoft Word document. Followed by that, coding was initiated through an inductive process of analysis using the constant comparative method [21]. The codes applied are keywords, which are used to categorize or organize text and are considered an essential part of qualitative research [22]. Because participants reflected on multiple ideas and perspectives, the data were analyzed for themes rather than mutually exclusive categories [23]. This process is "concerned with generating and plausibly suggesting (not provisionally testing) many properties and hypotheses about a general phenomenon" [21]. Two authors conducted the analysis independently. Each researcher examined the transcripts and discovered several codes, which they translated into themes that occurred repeatedly. The next stage involved interpreting the data by identifying any reoccurring themes throughout and highlighting any similarities and differences in the data. The final stage involved data verification. This process involves a process of checking the validity of understanding by rechecking the transcripts and codes again, thus allowing the researcher to verify the interpretation of the data arrived at previously [22].

Following the generation of themes, the authors examined the data as a team. The shared effort allowed the authors to triangulate the researchers' data, corroborate the results and improve the credibility [24]. The unit of analysis was the participants' perspectives towards the breadcrumbing behavior of their partners. The data was discussed by the researchers until they came to an agreement about the themes established, and this was reflected in the accounts. However, because the process of categorization was an inductive rather than deductive analysis, the kappa value was not calculated to establish the inter-rater reliability. Data were not confined into any qualitative framework (e.g., interdependence theory) to allow any themes to emerge, given that the purpose of the study was to explore a newer phenomenon rather than modifying an existing one.

**3. Results**

Based on the focused group discussions (see Figure 1), five themes emerged regarding conceptualizing breadcrumbing (RQ1), four themes for behaviors that constitute breadcrumbing (RQ2), four themes for impact of breadcrumbing on the breadcrumbie's health

and other relationships (RQ3), seven themes for breadcrumbing others and six themes for complying to breadcrumbing behaviors (RQ4) and five themes for effective ways of dealing with breadcrumbing (RQ5).

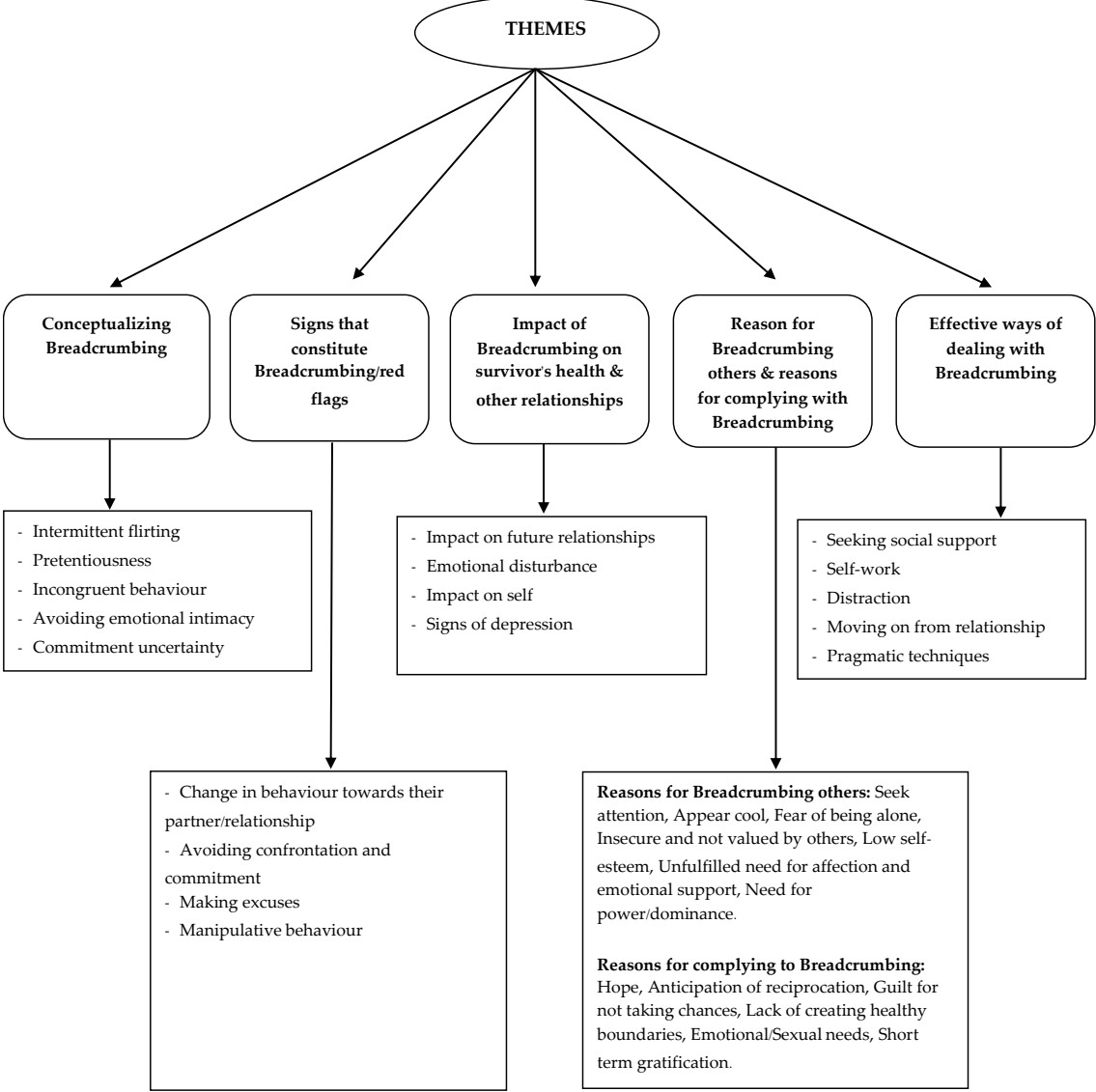

**Figure 1.** Themes of Breadcrumbing.

### 3.1. Conceptualizing Breadcrumbing

The key themes for conceptualizing breadcrumbing are intermittent flirtatious behavior, pretentiousness, incongruence, avoiding emotional intimacy and commitment. The first theme of intermittent flirtatious behavior constitutes of behaviors such as luring the person, flirting, giving compliments, being nice and showing romantic interest in the potential partner. Four out of nine participants (44%) stated experiencing flirting behavior in the initial stage of the relationship, where both show some romantic interest in each other.

The participants mentioned experiencing some pretentiousness from their partners as they moved forward in the relationship. Eight out of nine (89%) of the participants reported their partner showing pretentious behavior and giving them false hopes about the future of the relationship. Two out of nine participants (22%) reported that their partners said "I love you" way too early into the relationship, thus creating confusion in them. The partners shared physical and emotional intimacy that they both appreciated. This intimacy becomes a large factor for trusting the person and being hopeful about the future of the relationship.

One of the participants reported that her partner tried to convince her that he had changed as a person, and he admitted being wrong the last time just to gain sympathy and pretend to show a "nice image" in front of them.

The third theme is incongruence, where the breadcrumbie feels that something is wrong in their relationship, or things are not going well. They suspect their partner's real intentions about the relationship where they observe their partner's words and actions do not match. Four out of nine participants (44%) reported experiencing mixed signals from their partners. Four out of nine (44%) participants felt that their partners only sought physical intimacy or sexual satisfaction in the relationship. According to one participant (female, 23 y/o), "when you look the world through rose-coloured glasses, even the red flag looks like a normal flag. So, even if you want to, even if you do see, you kind of don't want to see because you think that, oh, man, right now I am happy, why I am destroying my own happiness." The red flags are evident at times, but the breadcrumbie chooses to ignore them in the hope of commitment from their partners.

The fourth theme is avoiding emotional intimacy, where the perpetrator eventually starts investing less time with their partner, acting indifferently towards their partners' needs and emotions and treating them like an option. In this theme, they exhibit very "hot and cold" kinds of behavior, making their partner feel a void in the relationship. The participants feel the lack of "genuineness" from their partners. In addition to that, online communication makes it easier for perpetrators to breadcrumb their partners compared to face-to face communication. In online platforms, it becomes convenient for the perpetrator to fake their emotions by using emoticons, hiding their personal life events and lying about them and having fewer things to offer in a relationship.

The final theme is commitment uncertainty, which is the essence of breadcrumbing. The perpetrator will try to put their best efforts to escape commitment and, henceforth, responsibility. They will portray behaviors such as making frequent excuses and delaying commitment, avoiding difficult conversations such as addressing the status of the relationship. They will keep their partners "on the hook" and not take things further. One of the participants mentioned (female, 24 y/o): "They are not willing to communicate, and then they make excuses and get out of it." At times, when the partner insists on them having a conversation about the relationship, chances are they might engage in potential gaslighting behavior. The perpetrator blames their partner for "always talking about their feelings" or they make it seem like it is their fault in seeking commitment. They also deny making any committal statements, planting self-doubt in their partner.

### 3.2. Signs/Behaviours That Constitute Breadcrumbing

The signs of breadcrumbing behavior are changes in behavior towards their partner, avoiding confrontation and commitment, making excuses and showing deceptive behavior. Among the nine participants, 56% of them stated that they noticed a sudden change in their partner's communication style and their overall behavior. Sixty-seven percent of them reported that they had experienced one-sidedness in the relationship. They saw that their partners are not putting enough effort into the relationship and it becomes emotionally draining for them. Two of the participants (22%) mentioned that their partners had a sudden change in their attitude towards the committed relationship. Two of them (22%) saw their partners distancing themselves from them.

As we discussed in the conceptualization of breadcrumbing, the essence of it lies in commitment uncertainty. When the two partners communicate about the nature of the relationship in the initial phase, they show a keen interest in each other. When things escalate, they start expecting some future with their partner. However, in this case, one of the partners avoids that conversation of "where are we?" by giving them false hopes or playing it cool. Two of the participants (22%) mentioned that their partners did not show any action towards the commitment of the relationship, whereas two of them (22%) stated that their partners avoided the discussion of the status of their relationship. To escape

the forthcoming conversation with the partner, the perpetrator makes some excuse that is emotional in nature, which inhibits any further confrontation from their partner.

The next theme is manipulative behavior, which suggests that the perpetrator employs different tactics to either getting their partner to do something or getting them to stop doing something. For instance, five out of nine participants (56%) stated that their partners were hiding their other sexual relationships from them. Four out of nine (44%) participants said they kept them on the hook by not taking things further or not ending things. This could be applied to fulfil their sexual or other psychogenic needs. We also noticed some potential gaslighting behaviors, such as making it seem like it is the breadcrumbie's fault for seeking commitment (22%); a decrease in responsibility or not taking responsibility for their actions; the denial of making any commitments; blaming their ex-partners for the failure of their relationships; not disclosing their real intentions of being in the relationship and making their partner feel insecure about themselves. In some instances, the perpetrator shows jealous behavior towards their partner (22%) and is possessive of them when they talk to other close friends.

### 3.3. Impact of Breadcrumbing on Breadcrumbie's Health and Other Relationship

The breadcrumbing impact on future relationships, fosters emotional disturbance, affects the self and is related to signs of depression. The most significant impact on the breadcrumbie's psychological state was not being able to trust other romantic partners in the future. Nine out of nine participants (100%) mentioned having trust issues in their future relationships. They reported having feelings of anxiety with their partners (67%), developing some sort of insecurity or feelings of unsafeness (56%), having difficulty in moving forward in the relationship, accepting commitment, or reciprocating it (33%), not able to open up emotionally or finding difficulty in expressing their emotions to their partners in the fear of not getting appreciated (22%) and questioning their own choices in relationships. One of the participants disclosed that she withdrew herself from society and her friends. She started spending less time with them, became less interactive than before and turned silent. One of them said that they also started breadcrumbing other people.

The second theme is emotional disturbance, which relates to a fear of repeating the same cycle in their romantic relationships (33%), feeling anger at oneself (22%) as well as numbing their emotions, overthinking and feeling frustration and jealousy. These emotions, if experienced for a long time, could contribute to their mental health and wellbeing on a significant level. The third theme revolves around a common factor, that is, self. Participants shared that they experienced a lot of self-doubt (33%), self-blaming (22%), loneliness, feeling of emptiness from within and feeling betrayed (22%). Their self-esteem took a nosedive because of the breadcrumbing experience (22%). They became desperate for relationships, but on the contrary, they became silent (22%). The triad of depression includes worthlessness, helplessness and hopelessness. In the case of breadcrumbing, one participant reported that she was diagnosed with clinical depression lasting for 6 months due to the breadcrumbing experience. She experienced low self-worth, helplessness and self-deprecating thoughts during that time. Two of the participants (22%) experienced a loss of appetite, contributing to psychosomatic factors. They reported experiencing exhaustion, loss of weight, weight gain and a loss of sleep. One of the participants reported dermatological effects, that is, she developed redness in her skin and acne for at least 4–5 months. She specified that upon confronting her partner about the leading on behavior and cutting him off from the relationship, she observed that her acne was gone. She explained that expressing her emotions to her partner helped her release the tension and suppressed emotions. This entails the magnitude of breadcrumbing in dating relationships.

### 3.4. Reason for Breadcrumbing Others and Reasons for Complying with Breadcrumbing

The major reasons for breadcrumbing were to seek attention, to appear cool, a fear of being alone, feeling insecure and not valued by peers, having low self-esteem, the need for affection and emotional support and the need for power/dominance. Five out of nine

participants (56%) believed that the perpetrator engages in breadcrumbing behavior to seek attention from their partners. Five out of nine (56%) of them posited that people want to appear cool in front of their friends or their social groups, which motivates them to engage in such behaviors. Three out of nine (33%) described a fear of being alone, whereas 22% of them said they feel insecure and that they are not valued by their peers. In addition, some participants believed that these individuals have low self-esteem, while others observed that they have some unfulfilled needs, such as the need for affection or emotional support (22%), the need for power, that is, to exhibit their power over their partners or to exert dominance and the need for validation and sexual needs.

Whereas reasons for complying with the breadcrumbing were found to be hope, the anticipation of reciprocation, feeling guilty for not taking chances, a lack of creating healthy boundaries, emotional, sexual and attentional needs and short-term gratification. For the reasons behind complying with breadcrumbing behavior, participants declared that hope is the biggest motivating factor for sticking to the person. Four out of nine participants (44%) said that they were hopeful about the future of the relationship, whereas two of them (22%) anticipated the reciprocation of feelings from their partners, which never happened. Two of them mentioned (22%) that they did not want to feel guilty for not taking any chances, while other participants stated that they did not want to feel guilty for not treating the person in the right way. One of them felt, in retrospect, that she lacked the ability to create healthy boundaries around her, which allowed the perpetrator to mislead her. In some instances, participants mentioned that they also have emotional, sexual or attentional needs, just like the perpetrator, so they enjoyed the short-term gratification and some extra love and attention from their partners.

### 3.5. Effective Ways of Dealing with Breadcrumbing

We asked the participants to describe their coping mechanism that differ across individuals. Moreover, we encouraged them to discuss the most effective ways of dealing with it based on their experiences. The effective coping strategies for the breadcrumbies are seeking social support, self-work, distracting oneself, moving on from the relationship and pragmatic techniques.

Seeking social support is a very important factor in people's lives to deal with problems and create strong connections with other people. This was the first theme gathered. Five out of nine participants (56%) sought social support, such as getting closer to their friends, family members and colleagues at work, whereas two of them (22%) even started sharing their experiences to other people through social media platforms. The second theme gathered was self-work and was seen as one of the most effective ways of dealing with breadcrumbing victimization. Nine out of nine participants (100%) opined that therapy is one of the best ways to understand their self and work on it. Four of them started investing time in themselves and doing things that brought them pleasure. Self-work also includes finding something meaningful, identifying unhealthy relationship patterns, acknowledging the importance of other relationships such as family and peers, giving importance to self and adapting a healthy approach in future relationships.

The third approach was distracting oneself by focusing on other aspects of life rather than on romantic relationships. Three out of nine participants (33%) tried to focus on their studies and career plans, whereas other participants mentioned that they focused on health, travel plans and workplace productivity. This helps them to take their minds off the relationship and push themselves into different events of life.

The most rational approach to deal with the breadcrumber is to cut off from the relationship. Three participants (33%) found it helpful to cut off from their partners to regain their strength and get out of the toxicity. Two of them (22%) confronted their partners to get the closure from the relationship. Getting closure is like starting a new journey from a fresh perspective. Expressing suppressed emotions to the perpetrators helps to clear out the negativity, a process which feels empowering and significantly contributes to our growth.

Some pragmatic techniques helped individuals overcome the effects of breadcrumbing. Four out of nine participants (44%) got into another relationship as a way to deal with their emotions. Two of them (22%) started breadcrumbing other individuals in dating. Two of them (22%) vented out their emotions through different means that was cathartic for them, while three of them (33%) confronted their partners face-to-face when they got a chance and communicated their experience to them with the hope that they will not repeat the same behavior with other people. One participant started smoking to deal with his emotions for a brief period. One participant believed in the karma factor, so being religious or adopting spiritual beliefs helped him to let go of his partner. One participant found physical exercise helpful, such as sports and other practices such as yoga and meditation.

## 4. Discussion

### 4.1. Conceptualization of Breadcrumbing

The primary aim of this study was to explore the phenomenon of breadcrumbing from a breadcrumbie's perspective and develop a conceptual framework of this behavior in the context of dating and relationships. Two focused group discussions were conducted with nine participants in total, resulting in a rich subjective analysis by the authors. The thematic analysis merged five themes of breadcrumbing behavior, along with the four other objectives of this study. The findings are discussed as below.

Breadcrumbing is the sexually luring behavior of leading on a romantic partner by providing false hopes and escaping accountability and commitment. In other words, the perpetrator emits mixed signals, oscillating between a showering of love and aloofness, creating confusion in the other's mind. The perpetrator displays love, affection and sexual interest to their partners at the beginning of dating, often posing to be charming, flirtatious and attractive. Although this behavior might seem to be common in a romantic relationship, it could be a subtle tactic used to lure someone just to obtain power and control over his or her partners. The participants described these individuals as "attractive, witty and charming" in the beginning of the relationship, but eventually unfolding into rather incongruent behavior. This can also be termed as "love bombing", a pop-culture term that indicates the excessive communication of a partner in controlling their partner's lives as a means of narcissistic self-enhancement [25]. However, the difference between flirtatious behavior and love bombing is to be noted. The distinction is very thin and yet important to understand. Love bombing fluctuates between radical affection and debasement depending on the perpetrator's needs.

The initial phase of the relationship can be challenging, especially in the age of the internet. Due to numerous dating apps in the market, the landscape of dating seems to be fundamentally changing. Dating apps offer user's an opportunity to evaluate potential partners that they are otherwise unlikely to encounter, allowing them to interact with others through computer-mediated communication and providing a mathematical algorithm to match with desirable partners [26]. Because of the easy access to dating apps and finding suitable partners, the chances of desiring an extra-dyadic relationship are high. The stronger the desire for the alternative partner, the more ambivalent the feelings towards the existing partner [27]. Another reason for extra-dyadic relationships, or incongruent behavior towards their partner, could be the unmet emotional needs of individuals [28]. According to the current results, a partner's intimacy and relationship satisfaction is inversely correlated with their unmet emotional needs. This is another possible explanation for individuals showing incongruent behavior (ambivalent) towards their partners.

Intimacy is a phenomenon that is sought-after and also feared. It is conceptualized as a sense of self-disclosure, the sharing of one's inner self and feeling close to one's partner [29–35]. It is a crucial part of a romantic relationship. The theme of avoiding intimacy is rooted in the individual differences concerning people's capacity to develop and maintain close relationships, and one of the factors is a fear of intimacy [36]. The fear of intimacy is rooted in the negative attitudes toward the self and others that develop early in life [37]. People are resistant to change these negative attitudes, comprising a

part of a person's identity and ultimately influencing their relationships. According to Firestone and Firetone [38], there is a strong association between a fear of intimacy and the desire for closeness. Therefore, individuals who engage in breadcrumbing behavior with their partners may desire less closeness due to their fear of intimacy. Behaviors such as spending less time than usual with their partners, not catering to their partner's needs and lack of genuineness may be associated with the fear that their partner will get too close to them. Another possible explanation for avoiding emotional intimacy is understood from the perspective of attachment styles. According to Bowlby [39,40], the beliefs and expectations—also known as *inner working models*—about the responsiveness of others in relation to the self is developed through an early interaction between infants and their caregivers. It means that people with secure attachment orientations have a tendency to engage in behaviors associated with intimacy compared to people with avoidant attachment styles. In the context of breadcrumbing behavior, it is likely that individuals who engage in such behaviors tend to have an avoidant attachment orientation. However, further research is needed to confirm this hypothesis.

Lastly, the theme of commitment uncertainty in the relationship is discussed. Commitment is the essence of what it means to be in an exclusive romantic relationship, driving how much partners invest, engage and identify with one another and the relationship. It is the fabric of how partners understand each other and develop their worldviews, making commitment an important fundamental aspect of human experience [41]. The authors explained this theme with the help of Stanley's *theory of commitment* that posits two overarching types of commitment, i.e., dedication and constraint [42]. It indicates that the partners who are able to meet their needs and derive a sense of meaning and purpose from their relationships do not seek alternative relationships. The dedication to develop a strong couple identity, have a long-term view in the relationship and the willingness to sacrifice may be less prevalent among the breadcrumbing perpetrators, resulting in incongruence and the fear of intimacy. Moreover, the other component of constraint in Stanley's theory is to be noted here. Individuals who generally have social pressures, financial obligations, child welfare or structural investments are encouraged to commit to their partners without a strong sense of dedication. However, in the case of breadcrumbing in dating, partners are not constrained by any such forces, thus making it easier for individuals to engage in breadcrumbing. Stanley et al. [42] explains that a lack of commitment and relationship dissolution is not the same. As commitment starts to waver, the uncertainty sets in and it influences the individual's cognition (e.g., questioning the status of the relationship), affective responses (e.g., intense emotional reactions to partner's behaviors) and behaviors (e.g., avoiding intimacy with the partner, spending time with them etc.). It is possible that the breadcrumbing perpetrator intends to form a sexual alliance disguised in a romantic courtship, but further research concerning perpetrators' perspectives need to be conducted.

### 4.2. Breadcrumbing as a Subtle Manipulation Tactic

Manipulation is defined as an intentional act used to influence, alter or shape their selected environments [43]. People use different manipulation tactics in social interactions. Especially in the context of dating, Buss et al. [44] have identified a few tactics, including charm, silent treatment, coercion, reason, regression and debasement as common tactics of manipulation. The participants in the current study described some of the behaviors that they considered as "red flags". We shall look into some of these red flags and associate it with the tactics given by Buss et al. [44]. One of the themes of "sudden change in the person's behaviour" constitutes behaviors such as a change in communication style towards their partner, distancing himself or herself and giving less attention, a change in attitude towards long-term relationship and so on. These sets of behaviors indicate ambiguity towards the partner and a dissatisfaction in the relationship. There may be a hidden need that is unfulfilled along with a lack of assertiveness to demand the same, which may result in a passive form of communication with their partners. Buss calls it a *silent treatment*, where the partner either ignores the partner or becomes silent and unresponsive to their needs—a

passive way of saying that they need something in terms of intimacy, affection, love or attention. Manipulation tactics are used for either behavioral instigation (getting them to do something) or behavioral termination (getting them to stop doing something) [43]. Charm or flirtatious behavior is used to fulfil their needs, a way to elicit a behavior from their partner, whereas silent treatment or coercion is used to stop their partner from doing something that they do not agree with.

Another red flag that was identified was "avoiding confrontation", or addressing relationship issues pertaining to their growth such as avoiding difficult conversations, refraining from taking steps for the future of the relationship, hiding personal information, not disclosing the purpose of the relationship and so on. This behavior of withholding personal information and feelings is a way of showing power and control in the relationship. It signifies that the relationship will operate on the perpetrator's terms, and if questioned or forced to change, they will try to alter the partner's choices either by using charm or silent treatment. Another explanation for avoiding confrontation is the unpleasant feelings attached to conflict and potentially deleterious outcome [45]. The perpetrator does not want to engage in difficult conversations that will reveal their true intentions, and eventually, they will lose control and power in the relationship. However, avoiding conflict is not necessarily a problem, but it becomes distressing to the other partner when it becomes an important issue to discuss. The participants reported anxiety, overthinking and ruminating over the unaddressed issue. More often than not, the perpetrator makes excuses to get out of the difficult conversation. The participants discussed that the excuses are "unquestionable, and even guilt-inducing". The tactic that perpetrators use to get out of such conversations is *reason* and *debasement*. Reason is a plausible explanation as to why one does not engage in a difficult conversation, whereas debasement is a tactic where the person devalues and lowers oneself to invoke a response from the other. In both cases, the other partner may feel inhibited to counter-question them, resulting in the avoidance of conflict. Another set of behaviors include lying to the partner, hiding other sexual relationships, diffusion of responsibility, denial, blaming the other person or being possessive of their partner and is associated with *hardball*. This is a tactic pertaining to violence, threatening or deceiving the other, where the perpetrator plays the role of an authority having power and control over the other person. This usually occurs during a conflict between the partners, especially when the conversation is not in favor of the perpetrator. There is another pop-culture term called *gaslighting*—a psychological violence occurring in intimate relationships, where one partner displays controlling behaviors toward the other to the extent that the other partner questions their own reality [46]. It is associated with behaviors such as the denial of abusive behaviors and presenting false information in a way that causes the other person to doubt his or her own perception or memory. The perpetrators often use tactics of aggression, lying, degradation, physical violence, emotional blackmail, deception and threatening the other. These behaviors are similar to the hardball tactic given by Buss. However, the perception of it being a red flag depends on the individuals. Further research is required to identify the personality correlates of perpetrators and the breadcrumbies of breadcrumbing to understand the impact on their other relationships.

*4.3. Psychological Impact on the Breadcrumbies*

Breadcrumbing, a subtle form of manipulation, has a temporary and yet strong impact on the breadcrumbies. The hot and cold behaviors of the perpetrators create a state of constant confusion and anxiety in the breadcrumbies. It is a slyly fabricated web of false hopes and lack of genuineness that leads the other person to have trust issues, anxiety, insecurity, shame, social withdrawal, frustration, helplessness, overthinking and a fear of repeating the cycle. The participants reported having experienced emotional disturbance for a brief period, resulting in symptoms of anxiety and depression. One of the participants had reported having the symptoms of acne and skin redness. Another participant reported having signs of major depression disorder (MDD), which was clinically diagnosed after 6 months of prolonged breadcrumbing behavior from their partner. Moreover, most of

the participants experienced loneliness, self-doubt, low self-esteem, low self-worth, self-deprecating thoughts and shyness. Further research needs to be conducted to understand the psychosomatic issues pertaining to breadcrumbing and the impact on self-concept and work productivity.

### 4.4. Effective Coping Strategies

Some effective coping strategies were collectively discussed and agreed upon by the participants. The first one is *seeking social support* including family support and support found from peers and in the workplace. Social support makes one feel less alone, helps them in processing emotions and makes them feel confident in making certain choices. The stronger the social support, the more secure the person feels [47]. Another study suggests that the absence of a satisfying social support is significantly related to poorer physical and psychological health outcomes [48]. Another coping strategy discussed is *distracting oneself* through shifting attention to more meaningful activities such as concentrating on physical health, studies, work, travel and other relationships. Engaging in meaningful work provides you with a sense of satisfaction and an improved quality of work-life [49]. Individuals can engage in *self-work* to move on from the relationship with the help of therapy, by investing more time in oneself, identifying their own unhealthy patterns in the relationship, giving self-importance and having gratitude for other relationships such as those of friends and family. This helps them understand their own behavior and the choices they make at every step in the relationship, making them self-aware and mindful of making future decisions. These are relatively healthier approaches to dealing with breadcrumbing. Some pragmatic techniques that the participants used were yoga and meditation, engaging in sports, venting out emotions, getting into another relationship and smoking. Further research is suggested to identify the most effective strategies for dealing with breadcrumbing.

### 4.5. Reason for Breadcrumbing Behaviour

Breadcrumbing, also known as "leading someone on", occurs in a romantic relationship when one person gives mixed signals or occasional attention to keep the other person interested in them, without actually committing to a relationship. This can be done for various reasons, such as wanting to keep options open, not being ready for a commitment, or enjoying the attention from multiple people [50]. It can be hurtful and manipulative and can leave the person being breadcrumbed feeling confused and insecure about the relationship. The participants of the study stated a few reasons behind breadcrumbing behavior, including wanting attention (*Participant NS: "he just wanted my attention whenever he felt alone or bored, but he never came to help me when I felt alone"*), to appear "cool" in front of others (*Participant DJ: "my boyfriend flaunts to his friends that he had sex with me, which is a big deal for them. It made me think that he just wants to tell his friends and show off"*), a fear of being alone, feeling insecure, not feeling valued, low self-esteem, sexual (*Participant PN: "every time he came to meet me, we either made out or got physical, but then slowly he distanced himself"*) or emotional need fulfillment, not being able to commit and not being able to open up emotionally. According to the article by Pattemore [50], breadcrumbers might have low self-esteem, insecure attachment style and personality traits such as narcissism. However, there is lack of evidence to this claim, so further studies need to confirm this hypothesis.

### 4.6. Limitations

A major limitation of this study is the lack of generalization of the results for the general population. The data were gathered from two focus group discussions consisting of a total of nine participants with the unequal ration of men and women (men = one, women = eight). Another limitation is that the data were collected only from the breadcrumbie's perspective, and data were not gathered from the breadcrumber's perspective. Therefore, in order to get a clear picture of the breadcrumbing phenomenon, the perpetrator's point

of view is important to study. Moreover, the study does not focus on the partners in committed relationships or marriage, which is another limitation of the study.

Additionally, qualitative researchers have pointed out that participants can gloss over event details, dramatize others, or refuse to give honest answers for different reasons, such as to protect themselves or someone else [51], although it does not mean that they are generally being dishonest. Qualitative research requires inquiring about participants' experiences of events that happen in their lives, and we have to assume as a limitation that participants are not always telling us "the whole truth" about their experiences. We also have to acknowledge that participants in our study could be aware of the situation suffered and could experience psychological discomfort sharing and confronting their own events and feelings with other participants' experiences during the group discussion. However, past research has shown that individuals sharing their life stories to an interested listener can experience positive and therapeutic effects from participation in qualitative studies [52,53].

### 4.7. Implications and Future Scope of Research

This study can be helpful in creating awareness among the individuals who form relationships through online platforms and other means regarding this behavior pattern. Individuals who frequently utilize dating apps to find meaningful relationships might fall into the trap of breadcrumbing. Along with that, mental health professionals must also be aware of this form of manipulative behavior that has a significant impact on an individual's health and wellbeing. They can help the clients experiencing breadcrumbing victimization through counselling and psychoeducation. This study can provide insights into the conflict resolution approach between couples. The personality characteristics of breadcrumbing perpetrators and victims could be identified with the help of this study, for it is important to educate individuals on how to form and maintain healthy relationships. This study can be helpful to the counsellors who deal with couples or marital issues.

The study can be extended by taking interviews of breadcrumbing perpetrators as well as the victims to understand both parties' perspectives to avoid any biases. With the help of the themes established in the current study, a scale for breadcrumbing victimization could be developed to identify the behaviors that are used against victims. It will be helpful for the market research of dating apps to take necessary actions to prevent such behaviors. Another suggestion would be to study the prevalence of breadcrumbing victimization in India and create workshops to educate individuals about this behavior pattern. Lastly, to validate the current research, various projective tests can be administered on perpetrators and victims to understand their needs, attachment pattern and personality.

### 5. Conclusions

Breadcrumbing can be conceptualized as five interconnected behavior patterns following in an order: (1) the individual displays intermittent flirting behavior to their partner to lure them, (2) the individual shows pretentious behavior, leading their partner into believing that the relationship holds a deeper meaning and commitment, (3) expressing mixed signals, where their actions and commitments are incongruent, (4) the individual avoids confrontation, emotional intimacy, and escapes responsibility of their actions and (5) the individual avoids actions pertaining to commitment and the growth of the relationship. This cluster of behaviors have a temporary but intense emotional disturbance on their partners. Therefore, it can be considered as a subtle form of manipulation and emotional abuse.

**Author Contributions:** Conceptualization, V.K.; methodology, V.K. & S.U.; formal analysis, V.K. & S.U.; investigation, V.K.; writing—original draft preparation, V.K & R.N.; writing—review and editing, V.K. & R.N; supervision, S.U. All authors have read and agreed to the published version of the manuscript.

**Funding:** This research received no external funding.

**Institutional Review Board Statement:** The study was conducted in accordance with the Declaration of Helsinki. The Maharaja Sayajirao University of Baroda, Vadodara, does not have an Institutional Review Board (IRB). However, the following ethical considerations were taken into account when conducting the current study: (1) the voluntary participation of interviewees who can discontinue participation at any time during the study; (2) no risks and monetary benefits were exposed to the participants; (3) data to be preserved with the researcher for minimum period of 2 years; (4) no post-study benefits would be shared with the participants; (5) participants may be informed about the study results if they wish to know; (6) data will be preserved with the utmost confidentiality; (7) no personal data/information of the participants will be shared; (8) video recording of the focused group discussions was conducted with the approval/consent of the participants; (9) the final study results may be published without revealing the personal identity of the participants; (10) a licensed clinical psychologist was there as a standby in case the participants experienced psychological distress during/after the focused group discussion.

**Informed Consent Statement:** Informed consent was obtained from all subjects involved in the study.

**Data Availability Statement:** The data that support the findings of this study are available on request from the corresponding author.

**Acknowledgments:** We thank Rashmin Sompura to guide us throughout the whole project. We also thank our colleagues and friends to support us.

**Conflicts of Interest:** The authors declare no conflict of interest.

## Appendix A. Focus Group Script

*Introduction:* Thank you all for coming today. My name is Vivek Khattar and I am a final year student earning my Master of Arts in Psychology from The Maharaja Sayajirao University of Baroda. I am working on my dissertation thesis on the topic of breadcumbing experiences among young adults in dating and relationships. The best way to do this is to talk to people who have first-hand experience with breadcrumbing. Therefore, I am holding this group discussion with you people where you can share your experiences, ideas and perspectives related to breadcrumbing. Please do not feel shy; we want to hear from all of you about your time here. There are no right or wrong answers. I simply want to hear your thoughts. I have some questions for you but also feel free to add other things you feel are important as we go along. During our discussion, I will be taking notes and to remind me if I forget to ask something, but since I cannot write down every word said, I would like to record the discussion so that I do not miss anything. Please do not be concerned about this; our discussion will stay confidential and only my supervisor and I will listen to the recording. Is it OK with everyone to record the discussion?

During our discussion, please let everyone share their views, but only one person should speak at a time so that the recording will be clear. Just join in when you have something to say, I will not be going around the group for every question. Remember we want to hear all your views. It's OK to disagree with others if you have a different experience or perspective but please also respect other people's views. Also, everything that you hear today should be confidential and not shared with people who are outside the group. This discussion will last between 60–90 min. Are there any questions before we start?

Let us start by introducing ourselves (rapport building).

Let us each share our first names, where are you from and what do you currently do.

Can you share your dating experience in today's digital world?

Which medium of communication do you prefer: offline or online? (probe: state reasons).

*Interview questions:*

1. Even though I had mentioned the definition of breadcrumbing, I would like to hear from you how you would define breadcrumbing. (Probe: the frequency/threshold)
2. Where did you meet and what was the medium of communication?

3. If you feel comfortable, can you please give examples of your personal experiences? (Probes: at what point of time in the relationship, what kind of hints)
4. According to you, what are the signs to identify BC?
5. Does it have any impact on your other relationships? (Probe: how)
6. Does breadcrumbing have a significant impact on your health-physical or mental? If yes, how? (How long did was the impact on your physical and mental health?
7. Why do you think people do it?
8. Why do you think people take it?
9. How did you deal with it?
10. What is the best way to deal with it?

Closing questions:

1. If your friend or a closed one were going through such experience, what would you do/advise them? What message will you give them?
2. What message would you like to give to those who breadcrumb other people on purpose?

Summarize the whole session and ask the group if it was an accurate reflection of the discussion.

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
