# Peer review of "Young Adults’ Perception of Breadcrumbing Victimization in Dating Relationships"

_societies, doi:10.3390/soc13020041_

Round 1

Reviewer 1 Report

The research problem undertaken by the author is interesting and important. The author tries to describe the phenomenon of breadcrumbing, its causes, manifestations and consequences. The methodological assumptions of the research are very well described. The research was conducted using focus interviews. The groups consist of few participants, but given that the study relates to very delicate and painful experiences, collecting data from a large number of people seems to be very difficult. For this reason, the research results are worth publishing. The research was not approved by the ethics committee, but the ethical dimension of the research was considered and described. I think two things should be corrected. First, limitations. The study concerned a very intimate sphere and this implied the issue of immaturity of the respondents themselves. They are the victims, but they allowed the perpetrators to manipulate them. This study may have made them, as victims, aware of this problem and caused psychological discomfort. Consequently, they could reveal only part of the truth about their experience. Secondly, in my opinion, giving data as a percentage makes no sense. Qualitative research is not about giving percentages, but writing about a phenomenon. Giving percentage data with such a small research sample seems pointless. In my opinion, instead of percentage data, more attention should be paid to the respondents' statements, the way they describe their experiences, so as to diagnose the reasons for their consent to manipulation and harm.

Author Response

First, we are extremely grateful for the comments from the reviewers, which we believe have contributed to significant improvements in the paper.

We have commented on each piece of feedback separately and we have attached an updated version of our paper after revisions.

Changes introduced in the manuscript haven been tracked.

Reviewer 1

Comment 1: First, limitations. The study concerned a very intimate sphere and this implied the issue of immaturity of the respondents themselves. They are the victims, but they allowed the perpetrators to manipulate them. This study may have made them, as victims, aware of this problem and caused psychological discomfort. Consequently, they could reveal only part of the truth about their experience.

Authors’ response: thank you very much for your kind review and to point out some relevant studies related with our manuscript. We have modified the limitations section accordingly pointing out the limitations raised by the reviewer. We have added the following paragraph (page 13-14):

“Additionally, qualitative researchers have pointed out that participants can glossing over events details, dramatizing others, or refusing to give honest answers for different reasons such as protect themselves or someone else 51, although it does not mean that they are being generally dishonest. Qualitative research requires inquiring participants about their experiences of events that happen in their lives, and we have to assume as a limitation that participants are not always telling us “the whole truth” about their experiences. We have also to acknowledge that participants in our study could be aware of the situation suffered and could experience psychological discomfort sharing and confronting their own events and feelings with other participants’ experiences during the group discussion. However, past research has shown that individuals sharing their life stories to an interested listener can experience positive and therapeutic effects from participation in qualitative studies 52,53.”

Comment 2: Secondly, in my opinion, giving data as a percentage makes no sense. Qualitative research is not about giving percentages but writing about a phenomenon. Giving percentage data with such a small research sample seems pointless. In my opinion, instead of percentage data, more attention should be paid to the respondents' statements, the way they describe their experiences, so as to diagnose the reasons for their consent to manipulation and harm.

Authors’ response: thank you. We have decided to keep the percentages as we believe is informative of how the participants introduce similar themes in the group discussion. However, following your advice we have introduced additional participants’ quotes to illustrate why breadcrumbing occurs, introducing more information about their experiences to this regard. You can see this new information in page 13.

Reviewer 2 Report

Dear authors, thank you for giving me the opportunity to review your manuscript: “Young Adults’ Perception of Breadcrumbing Victimization in Dating Relationships”. 

The article is clear, and I think this manuscript can contribute to the literature.

I send below some comments to improve the manuscript.

Introduction:

-       The introduction is relatively clear, but there is a need to explore further the results presented by the other studies on the topic.

Data Analysis

-        Did the authors calculate the Kappa value - fidelity index between the two authors that conducted the analysis independently.)?

Discussion

-       Please add some discussion about why the Breadcrumbing occurs.

Author Response

First, we are extremely grateful for the comments from the reviewers, which we believe have contributed to significant improvements in the paper.

We have commented on each piece of feedback separately and we have attached an updated version of our paper after revisions.

Changes introduced in the manuscript haven been tracked.

Reviewer 2

Comment 1: The introduction is relatively clear, but there is a need to explore further the results presented by the other studies on the topic.

Authors’ response: thank you for this suggestion. We have made several edits to improve description of the previous research on breadcrumbing. Please, see edits in page 2.

Comment 2: Did the authors calculate the Kappa value - fidelity index between the two authors that conducted the analysis independently.)?

Authors’ response: thank you for this commentary. We did not calculate the Kappa value given that inter-rater reliability is most appropriate for the use of deductive rather than inductive coding systems. Such deductive approaches rely on the application of pre-determined coding systems that are typically based on manifest, rather than latent content. Given that our study was exploratory and there is not previous qualitative research on breadcrumbing, we think that an inductive analysis was more appropriate.  Nonetheless, we have indicated that we did not calculate the kappa value in the manuscript. See page 5, lines 198-199.

Comment 3: Please add some discussion about why the Breadcrumbing occurs.

Authors’ response: thank you for this suggestion. The inclusion and exclusion criteria have been described at the beginning of the participants subsection. We have added a new sub-section in the discussion to describe why breadcrumbing occurs including new quotes from participants to this regard (see page 13).